# DSPNet: Towards Slimmable Pretrained Networks based on Discriminative Self-supervised Learning

## Abstract

Self-supervised learning (SSL) has achieved promising downstream performance. However, when facing various resource budgets in real-world applications, it costs a huge computation burden to pretrain multiple networks of various sizes one by one. In this paper, we propose **D**iscriminative-SSL-based **S**limmable **P**retrained **Net**works (**DSPNet**), which can be trained once and then slimmed to multiple sub-networks of various sizes, each of which faithfully learns good representation and can serve as good initialization for downstream tasks with various resource budgets. Specifically, we extend the idea of slimmable networks to a discriminative SSL paradigm, by integrating SSL and knowledge distillation gracefully. We show comparable or improved performance of DSPNet on ImageNet to the networks individually pretrained one by one under the linear evaluation and semi-supervised evaluation protocols, while reducing large training cost. The pretrained models also generalize well on downstream detection and segmentation tasks. Code will be made public.

## 1 Introduction

Recently, self-supervised learning (SSL) draws much attention to researchers where good representations are learned without dependencies on manual annotations. Such representations are considered to suffer less human bias and enjoy better transferability to downstream tasks. Generally, SSL solves a well-designed pretext task, such as image colorization (Zhang et al., 2016), jigsaw puzzle solving (Noroozi & Favaro, 2016), instance discrimination (Dosovitskiy et al., 2014), and masked image modeling (Bao et al., 2021). According to the different types of pretext tasks, SSL methods can be categorized into generative approaches and discriminative approaches. Discriminative approaches received more interest during the past few years, especially the ones following the instance discrimination pretext tasks (He et al., 2020; Chen et al., 2020a; Grill et al., 2020; Chen et al., 2020b; Caron et al., 2020; Chen & He, 2021). These pretraining approaches have shown superiority over their ImageNet-supervised counterpart in multiple downstream tasks. However, the pretraining is time-consuming and costs large computing resources, *e.g.*, the pretraining of BYOL costs over 4000 TPU hours on the Cloud TPU v3 cores.

In real-world applications, the resource budgets vary in a wide range for the practical deployment of deep networks. A single trained network cannot achieve optimal accuracy-efficiency trade-offs across different devices. It means that we usually need multiple networks of different sizes. A naive solution to apply pretraining and also meet such conditions is to pretrain them one by one and then fine-tune them to accomplish specific downstream tasks. However, their pretraining cost grows approximately linearly as the number of desired networks increases. It costs so many computing resources that this naive solution is far from being a practical way to make use of the favorable self-supervised pretraining.

In this paper, a feasible approach is proposed to address this problem - developing slimmable pretrained networks that can be trained once and then slimmed to multiple sub-networks of different sizes, each of which learns good representation and can serve as good initialization for downstream tasks with various resource budgets. To this end, we take inspiration from the idea of slimmable networks (Yu et al., 2018), which can be trained once and executed at different scale, and permit

the network FLOPs to be dynamically configurable at runtime. Nonetheless, slimmable networks (Yu et al., 2018) and the subsequent works (Yu & Huang, 2019; Yu et al., 2020; Cai et al., 2019; Yang et al., 2020) all focus on specific tasks with supervision from manual annotations, while we concentrate on SSL-based representation learning without manual annotations. To bridge this gap, we build our approach upon BYOL (Grill et al., 2020), a representative discriminative SSL method. Specifically, as multiple networks of different sizes with good representations need to be pretrained at one go, we thus construct them within a network family that are built with the same basic building blocks but with different widths and depths. We use *desired networks* (DNs) to denote them. Different from the original design in BYOL that the online and target networks share the same architecture at each training iteration, our online network is specifically deployed by activating the randomly sampled sub-networks in it during pretraining, which include all the above *desired networks*. We also perform BYOL's similarity loss between the target branch and the sampled sub-networks in the online branch, and the target network is also updated by exponential moving averages of the online network. After pretraining, DNs can be slimmed from the online network and each of them learns good representation. In this way, DNs are pretrained together by sharing weights, which can reduce large training cost compared with pretraining them one by one. We name the proposed approach as **D**iscriminative-SSL-based **S**limmable **P**retrained **N**etworks (**DSPNet**). Our contributions are summarized as follows:

- We propose DSPNet, which can be pretrained once with SSL and then slimmed to multiple sub-networks of various sizes, each of which learns good representation and serves as good initialization for downstream tasks with various resource budgets.
- We show that our slimmed pretrained networks achieve comparable or improved performance to individually pretrained ones under various evaluation protocols, while large training cost is reduced.
- With extensive experiments, we show that DSPNet also performs on par or better than previous distillation-based SSL methods, which can only obtain a single network with good representation by once training.

## 2 RELATED WORK

**Self-supervised Learning.** Recent self-supervised learning (SSL) approaches have shown prominent results by pretraining the models on ImageNet (Deng et al., 2009) and transferring them to downstream tasks. It largely reduces the performance gap or even surpasses with respect to the supervised models, especially when adopting large encoders. Generally, SSL solves a well-designed pretext task. According to the types of pretext tasks, SSL methods can be categorized into generative approaches and discriminative approaches. Generative approaches focus on reconstructing original data (Pathak et al., 2016; Bao et al., 2021; He et al., 2021; Zhang et al., 2016). Discriminative approaches have drawn much attention in recent years, especially those based on the instance discrimination pretext tasks (Dosovitskiy et al., 2014), which consider each image in a dataset as its own class. Among them, contrastive learning (Hadsell et al., 2006) methods achieve more promising performance, where the representations of different views of the same image are brought closer (positive pairs) while the representations of views from different images (negative pairs) are pushed apart. BYOL (Grill et al., 2020) further gets rid of negative pairs while preserving high performance, namely non-contrastive method. Most of the above methods rely on large batch size and long training schedule, which cost large computing resources.

**Slimmable Networks.** Slimmable networks are a class of networks executable at different scales, which permit the network FLOPs to be dynamically configurable at runtime and enable the customers to trade off between accuracy and latency while deploying deep networks. The original version (Yu et al., 2018) achieves networks slimmable at different widths, and US-Nets (Yu & Huang, 2019) further extends slimmable networks to be executed at arbitrary widths, and also proposes two improved training techniques, namely the *sandwich rule* and *inplace distillation*. The subsequent works go beyond only changing the network width, *e.g.*, additionally changing the network depth (Cai et al., 2019), kernel size (Yu et al., 2020) and input resolutions (Yang et al., 2020). However, previous works all focus on specific tasks with supervision from manual annotations. Our approach extends them to SSL-based representation learning, to obtain slimmable pretrained models by once training.

**Self-supervised Learning with Distillation.** Some research focuses on applying the aforementioned self-supervised methods to train lightweight networks, while the experimental results in Fang et al. (2020) show that the naive SSL methods do not work well for lightweight networks. To address this problem, knowledge distillation (Hinton et al., 2015) is adopted by introducing a larger self-supervised trained teacher to help the lightweight networks learn good representations. For instance, CompRess (Abbasi Koohpayegani et al., 2020) and SEED (Fang et al., 2020) employ knowledge distillation to improve the self-supervised visual representation learning capability of small models, relying on MoCo (Chen et al., 2020b) framework. A more flexible framework is proposed in Gao et al. (2021) with contrastive learning adopted to transfer teachers' knowledge to smaller students. However, these two-stage methods all rely on the pretrained teachers and keep them frozen during distilling, which may be tedious and inefficient. Recently, some methods adopt online distillation (Bhat et al., 2021; Choi et al., 2021), which recommends training the teacher and student at the same time. Our approach bears some similarity to them in that we also conduct SSL and knowledge distillation simultaneously. However, our method is aimed to obtain multiple networks of different sizes by training once to meet the demand of the practical deployment of deep networks under various resource budgets. Moreover, it can reduce large training cost.

## 3 PRELIMINARIES

In this section, the representative discriminative SSL methods BYOL (Grill et al., 2020) and slimmable networks are firstly summarized as preliminaries for our method.

### 3.1 BYOL

BYOL Grill et al. (2020) consists of two paralleled branches, namely online and target branches. Given an image $x$ from datasets, two augmented views $v$ and $v'$ are obtained by applying different image augmentations respectively. From the first view $v$, the online network parameterized by $\theta$ first outputs a representation $y_\theta \triangleq f_\theta(v)$ and then a projection $z_\theta \triangleq g_\theta(y)$, and the target network parameterized by $\xi$ outputs a representation $y'_\xi \triangleq f_\xi(v')$ and the projection $z'_\xi \triangleq g_\xi(y')$ from the second view $v'$. A predictor $q_\chi$ parameterized by $\chi$ is applied to the online branch, and mean squared error is adopted as prediction loss:

$$L_{\theta,\xi,\chi} = \|\bar{q}_\chi(z_\theta) - \bar{z}'_\xi\|_2^2 \,, \tag{1}$$

where $\bar{q}_\chi(z_\theta) \triangleq q_\chi(z_\theta)/\|q_\chi(z_\theta)\|_2$ and $\bar{z}'_\xi \triangleq z'_\xi/\|z'_\xi\|_2$ are the normalized version. Usually, a symmetric loss $\widetilde{L}_{\theta,\xi,\chi}$ is also computed by separately feeding $v'$ to the online network and $v$ to the target network. The parameters of the online branch $\theta$ and $\chi$ are finally updated by backpropagation from the total loss $L_{\text{BYOL}} = L_{\theta,\xi,\chi} + \widetilde{L}_{\theta,\xi,\chi}$, and the parameters of the target branch $\xi$ are updated as an exponential moving average (EMA) of $\theta$.

### 3.2 SLIMMABLE NETWORKS

Slimmable networks (Yu et al., 2018) are a class of networks that can be executable at different scales. As shown in Fig. 1, by dropping specific blocks or inactivating parts of channels, one network can be slimmed to smaller sub-networks. Usually, a single trained network can only be deployed at the full scale, while slimmable networks can be deployed as any one of the pre-defined sub-networks to accomplish specific tasks. To train slimmable networks, a practical and cost-effective strategy is proposed, namely *sandwich rule*. During training, only the smallest, largest and a few randomly sampled sub-networks are used to calculate the loss in each iteration. Besides, *inplace distillation* (Yu & Huang, 2019) is also a useful training technique, in which the knowledge inside the largest network is transferred to sub-networks by simply adding distillation loss between them.

## 4 METHOD

Our goal is to obtain multiple networks of various sizes within a network family by training once with SSL, all of which have good representations after optimizing. We achieve it by integrating

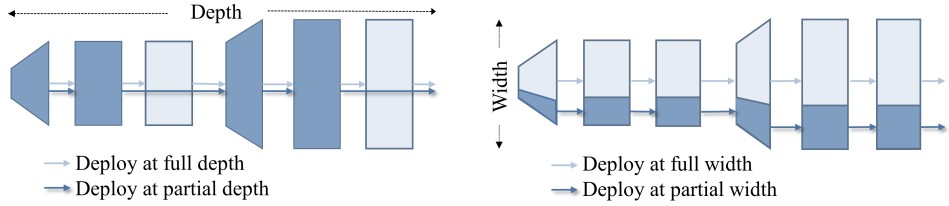

Figure 1: By dropping specific blocks *(Left)* and inactivating parts of channels *(Right)*, one network can be slimmed to a smaller sub-network. We show them separately for clarity, while they could be applied together.

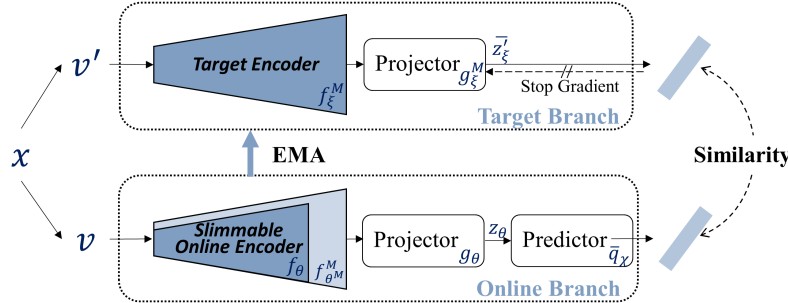

Discriminative-SSL-based Slimmable Pretrained Networks (**DSPNet**)

Figure 2: **Architecture of our DSPNet**. Based on BYOL (Grill et al., 2020), our online encoder is deployed by activating the randomly sampled sub-networks in it at each training iteration, while the target encoder is always deployed at full depth and width. After pretraining, the optimized online encoder can be slimmed to multiple networks of different sizes, and all of them have good representations.

BYOL (Grill et al., 2020) and slimmable networks (Yu et al., 2018). In this section, we introduce the architecture and training procedure of our **D**iscriminative-SSL-based **S**limmable **P**retrained **Net**works (**DSPNet**).

## 4.1 DSPNET

The overall architecture of our approach, namely DSPNet, is shown in Fig. 2. It largely follows BYOL (Grill et al., 2020), except for the encoder of the online branch. Unlike the original design in BYOL, in which the target encoder itself is the desired network with good representations after training, our desired networks are several slices of the online encoder[1]. They are sub-networks by removing specific channels and blocks from the whole online network, referred to as *desired networks* (DNs) in this paper.

To train DSPNet, we also adopt the *sandwich rule* (Yu et al., 2018). Specifically, at each training iteration, we sample $n$ sub-networks, involving the smallest, largest and randomly sampled ($n$-2) sub-networks, then calculate the loss for each of them, and finally apply gradients back-propagated from the accumulated loss of them. We illustrate the construction of the online encoder and its behavior during pretraining in Fig. 3. The projector and predictor along with the loss function all follow the original design in BYOL (Grill et al., 2020), except for the first linear layer in the projector of the online branch, because it should have an alterable input channel to adapt to the output representations with different dimensions from the online encoder. As for the target branch, the encoder has the same architecture as the whole online network and is always deployed at full width (*i.e.*, all channels are activated) and depth (*i.e.*, all blocks are maintained).

More specifically, for one sampled online network, which is defined by a set of weights $\theta$, we denote its encoder as $f_\theta \sim \mathcal{F}$, and then the projection of the representation from the online encoder is calculated as

$$z_\theta \triangleq g_\theta(f_\theta(v)) \tag{2}$$

---

[1]Since the target network is obtained by exponential moving average of the online network, it should converge to the same status as the online encoder. In our experiments, we find that the representations of the sliced networks from the target and online branches can achieve comparable results.

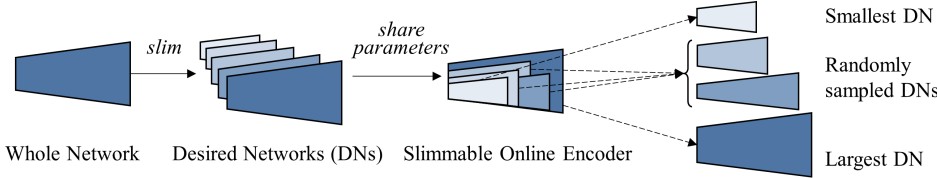

Figure 3: Construction of the online encoder and its behavior during pretraining in DSPNet.

for the first view $v$, where $g_\theta$ is the projector. It is noteworthy that if the whole online encoder is defined as $f^M$ parameterized by $\theta^M$, then the parameters of any sub-network are a subset of $\theta^M$. For the second view $v'$, the projection of the representation from the encoder of the target branch is calculated as

$$z'_\xi \triangleq g^M_\xi(f^M_\xi(v')), \tag{3}$$

where $f^M_\xi$ and $g^M_\xi$ are the encoder and projector of the target branch parameterized by $\xi$, which are deployed at full size the same as the whole online network, *i.e.*, $|\xi| = |\theta^M|$. Then, our prediction loss is defined as

$$L_{\theta,\xi,\chi} = \|\bar{q}_\chi(z_\theta) - \bar{z}'_\xi\|^2_2 , \tag{4}$$

where $q_\chi$ is the predictor parameterized by $\chi$ in the online branch, and $\bar{q}_\chi(z_\theta)$ and $\bar{z}'_\xi$ are the normalized version of $q_\chi(z_\theta)$ and $z'_\xi$.

We compute the loss by taking an unweighted sum of all training losses of sampled networks:

$$L_{\Theta,\xi,\chi} = \sum_{i=1}^{n} \|\bar{q}_\chi(z_{\theta_i}) - \bar{z}'_\xi\|^2_2 , \tag{5}$$

where $\Theta = \theta_1 \bigcup \theta_2 \bigcup \cdots \bigcup \theta_n$ and $\theta_i$ is the parameters of the $i$-th sampled online network. Since the largest sub-network (the whole one) is always sampled in the online branch following the *sandwich rule*, we can find that $\Theta = \theta^M$. We also symmetrize the loss in Eq. (5) by interchanging $v$ and $v'$ to compute $\widetilde{L}_{\theta^M,\xi,\chi}$ following the practice in BYOL. The total loss is $L^{\mathrm{DSPNet}}_{\theta^M,\xi,\chi} = L_{\theta^M,\xi,\chi} + \widetilde{L}_{\theta^M,\xi,\chi}$. $\theta^M$ along with $\chi$ are updated based on the gradients from backpropagation, and $\xi$ is updated by a momentum-based moving average of $\theta^M$ with hyper-parameter $\tau$, *i.e.*,

$$\theta^M, \chi \leftarrow \mathrm{optimizer}(\theta^M, \chi, \nabla_{\theta^M,\chi} L^{\mathrm{DSPNet}}_{\theta^M,\xi,\chi}) , \tag{6}$$

$$\xi \leftarrow \tau\xi + (1-\tau)\theta^M . \tag{7}$$

Since we can accumulate back-propagated gradients of all sampled networks and do not need to always keep all intermediate results, the training procedure is memory-efficient, with no more GPU memory cost than pretraining individual networks. Moreover, at each training iteration, $\bar{z}'_\xi$ is shared for all sampled online encoders, which means it is also time-efficient. After pretraining, the *desired networks* (DNs) can be slimmed from the whole online encoder, all of which are expected to have good representations.

## 4.2 BEHAVIORS BEHIND THE TRAINING OF DSPNET

Overall, our online encoder varies within the predefined DNs during pretraining. When the whole network is sampled, our training behaves like BYOL (Grill et al., 2020). It is essential for our approach to achieve good representations. While for the other sampled sub-networks, the supervision is built upon the similarity between the representations from the target branch (with the same size as the whole online network) and this partially activated online network, which can be viewed as knowledge distillation so that the representation capability is transferred and compressed to the sub-networks. It is noteworthy that, different from the *inplace distillation* of slimmable networks (Yu & Huang, 2019), our teacher is the target network rather than the whole online network, which is a momentum-based moving average of the whole online network and severed as a stable and reliable teacher. In this view, our training is jointly optimized towards self-supervised learning and knowledge distillation, and thus contributes to simultaneous representation learning and model compression.

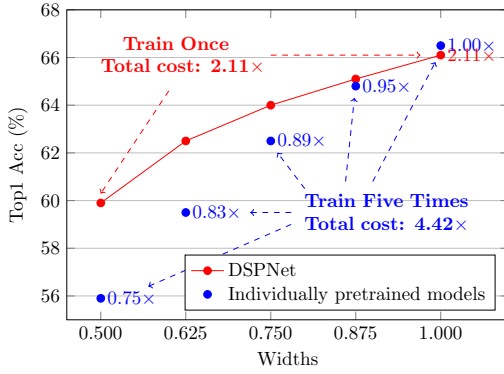 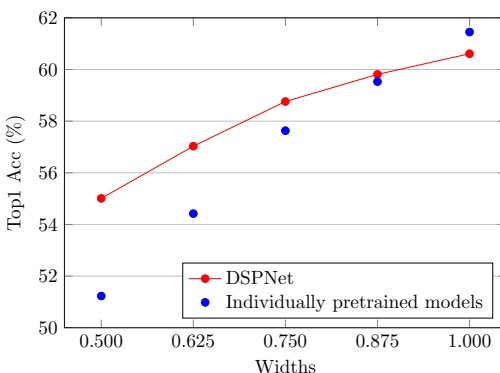

Figure 4: **Top1 accuracy (%) on ImageNet under linear evaluation** for DNs slimmed from our DSPNet and individually pretrained models. Relative training time is also reported.

Figure 5: **Semi-supervised learning on ImageNet** using 10% training examples for DNs slimmed from our DSPNet and individually pretrained models. Top1 accuracy (%) is reported.

## 5 EXPERIMENTS

We train our DSPNet on ImagetNet ILSVRC-2012 (Deng et al., 2009) without labels. Then we evaluate the representations of our obtained DNs on the validation set of ImageNet, including linear evaluation protocol and semi-supervised evaluation protocol. The transferability to the downstream tasks, such as object detection and instance segmentation on COCO (Lin et al., 2014), is also investigated to study the generalization.

### 5.1 COMPARISONS WITH INDIVIDUALLY PRETRAINED NETWORKS

One of the advantages of our method is that we can obtain multiple *desired networks* (DNs) by training once, which is cost-efficient compared with individually pretraining them one by one. Thus, experimental comparisons with individually pretrained networks are carried out in this section.

**Implementation Details.** We mainly follow the settings in BYOL (Grill et al., 2020). The same set of image augmentations as in BYOL (Grill et al., 2020) is adopted. Our DNs are MobileNet v3 (Howard et al., 2019) with different widths, *i.e.*, $[0.5, 0.625, 0.75, 0.875, 1.0]\times$. The projector and predictor are the same as in BYOL (Grill et al., 2020). We also adopt momentum BN (Li et al., 2021) for further accelerating. We use the LARS (You et al., 2017) optimizer with a cosine decay learning rate schedule (Loshchilov & Hutter, 2016) and 10-epoch warm-up period. We set the base learning rate to 0.2, scaled linearly (Goyal et al., 2017) with the batch size (*i.e.*, $0.2\times$BatchSize/256). The weight decay is $1.5 \cdot 10^{-6}$. The EMA parameter $\tau$ starts from $\tau_{base} = 0.996$ and is gradually increased to $1.0$ during training. We train for 300 epochs with a mini-batch size of 4096. The number of sampled sub-networks per iteration $n$ is 4. Besides, we constrain the online encoder to be deployed at full scale at the first 10 warm-up epochs, for robust training. As our baselines, we individually pretrain these models (MobileNet v3 at widths $[0.5, 0.625, 0.75, 0.875, 1.0]\times$) with BYOL (Grill et al., 2020). The same training recipe as ours is adopted.

**Linear Evaluation on ImageNet.** Following the ImageNet (Deng et al., 2009) linear evaluation protocol (Grill et al., 2020) as detailed in Appendix A.2, we train linear classifiers on top of the frozen representations with the full training set, and then report the top1 accuracy on the validation set. Note that the classifier is trained individually for each DN. As shown in Fig. 4, our method achieves comparable performance to the individually pretrained models. Specifically, for smaller DNs, our method achieves minor improvement, while showing slightly inferior results only for the largest one.

Furthermore, to evaluate the efficiency of our method, we report the relative pretraining time w.r.t. the individually pretrained MobileNet v3 at width $1\times$, which is also shown in Fig. 4. It is evaluated on 16 V100 32G GPUs. The pretraining of our DSPNet spends $2.11\times$ time w.r.t. the above baseline,

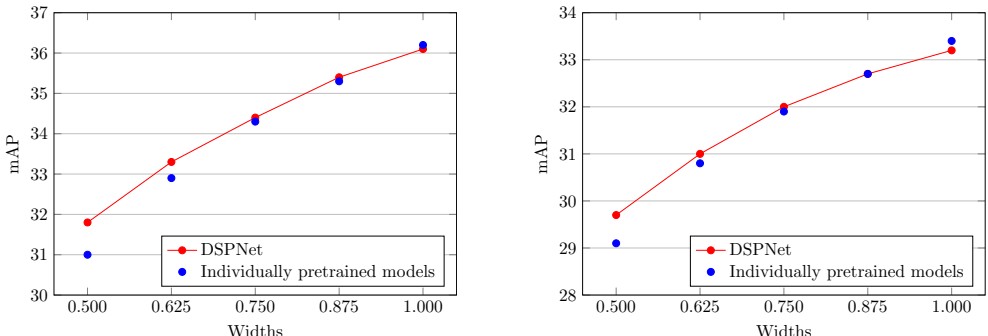

Figure 6: **COCO object detection** *(Left)* **and segmentation** *(Right)* based on Mask R-CNN. mAP is reported for both DNs slimmed from our DSPNet and individually pretrained models.

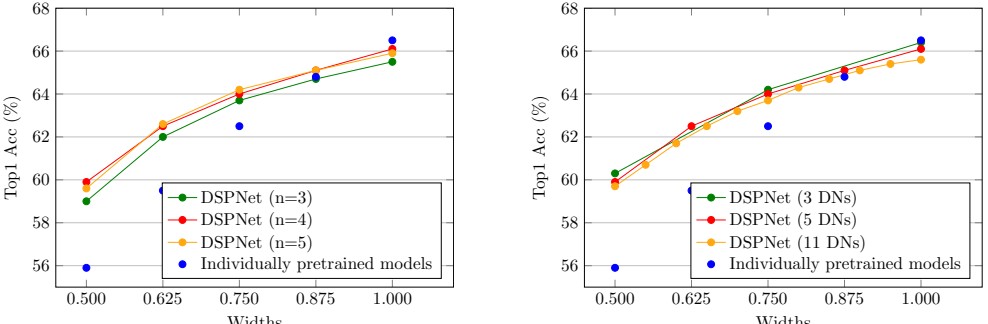

Figure 7: **Ablation study** under linear evaluation. We examine the effect of the number of sampled sub-networks per iteration *(Left)*, and the number of pre-defined DNs *(Right)*.

while it totally costs $4.42\times$ time to pretrain all the DNs one by one. About a half of training cost can be reduced by our method.

**Semi-supervised Evaluation on ImageNet.** We also evaluate the representations on a classification task with a small subset of ImageNet's training set available for fine-tuning them, *i.e.*, semi-supervised evaluation, as shown in Fig. 5. We fine-tune the pretrained models by only utilizing 10% of the labeled training data, including both multiple DNs slimmed from our DSPNet and individually pretrained models. Similar phenomena are observed as in linear evaluation. Higher accuracy than individually pretrained models is achieved except for the largest DN. More details can be found in Appendix A.3.

**Transferability to Other Downstream Tasks.** We use Mask R-CNN (He et al., 2017) with FPN (Lin et al., 2017) to examine whether our representations generalize well beyond classification tasks (see Appendix A.4 for more details). We fine-tune on COCO *train2017* set and evaluate on *val2017* set. All the model parameters are learnable. We follow the typical $2\times$ training strategy (Wu et al., 2019). As shown in Fig. 6, our bounding box AP and mask AP both outperform the individually pretrained counterparts for most models.

## 5.2 ABLATION STUDY

We now explore the effects of the hyper-parameters of our method. For concise ablations, all experiments are conducted based on MobileNet v3 (Howard et al., 2019) and the pretrained models are evaluated under the linear evaluation protocol.

**Number of Sampled Sub-networks Per Iteration** $n$**.** We first study the number of sampled sub-networks per training iteration. It is worth noting that larger $n$ leads to more training time. We train three kinds of DSPNet models with $n$ equal to 3, 4, or 5. As shown in the left of Fig. 7, the models pretrained with $n = 4$ achieve better results than $n = 3$ under linear evaluation, but further increasing to $n = 5$ does not lead to significant improvement. Thus, we adopt $n = 4$ in all our experiments by default.

**Number of DNs.**   We then investigate the effect of the number of DNs. The more DNs, the more pretrained models can be obtained by once training. As shown in the right of Fig. 7, we have pretrained 3-switch, 5-switch, and 11-switch DSPNet. They all show similar performance, demonstrating the scalability of our method.

### 5.3   Applying to Other Networks.

Our method is generally applicable to most representative network architectures. We further evaluate our method on ResNet (He et al., 2016). We construct several DNs based on the *BasicBlock* building blocks. Specifically, the channel numbers for the 4 stages are sampled from $\{[64, 128, 256, 512],\ [80, 160, 320, 640],$ $[96, 192, 384, 768]\}$ and the block numbers for the 4 stages are sampled from $\{[2, 2, 2, 2],\ [2, 3, 4, 3],$ $[3, 4, 6, 3]\}$, resulting in 9 networks as DNs. The smallest one among them is the well-known ResNet18, and the largest one is ResNet34 at width $1.5\times$. The top1 accuracy of linear evaluation w.r.t. their parameter numbers is shown in Fig. 8. The performance of these networks

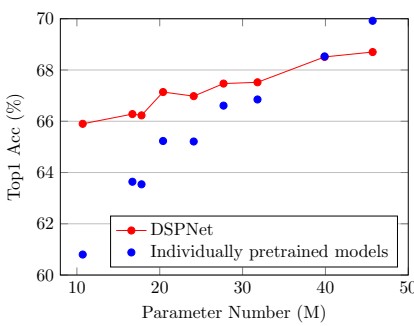

Figure 8: **Applying DSPNet to ResNet.**

individually pretrained with BYOL (Grill et al., 2020) is also evaluated. The same training configurations as those of MobileNet v3 are adopted. It can be drawn that for most smaller models, our DNs slimmed from the whole network outperforms the individually pretrained ones, while falling behind for a few larger ones. We attribute it to the conjecture that the whole online network makes a compromise between slimmablity and good representation.

### 5.4   Comparisons with SSL Methods with Distillation

We notice that there are some works (Fang et al., 2020; Abbasi Koohpayegani et al., 2020; Gao et al., 2021) focusing on the self-supervised training of lightweight networks with knowledge distillation. From some perspectives, our method can be viewed as an online distillation method by training the teacher (the target network) and students (sub-networks of the online network) simultaneously. Thus we compare our method with them by only picking the smallest DN slimmed from our DSPNet. We directly use the models introduced in Sec. 5.3, where the smallest DN is ResNet18.

First, we perform comparisons under the k-nearest-neighbor (kNN) classification and linear evaluation on ImageNet (Deng et al., 2009). As shown in Tab. 1a, we achieve better results than other distillation-based SSL methods (Fang et al., 2020; Abbasi Koohpayegani et al., 2020; Gao et al., 2021). It is worth noting that our method does not rely on well-pretrained teachers, and additional models with good representations can also be obtained at the same time.

Second, we evaluate the pretrained models under semi-supervised evaluation. Following previous works (Fang et al., 2020), we train a linear classifier on top of the frozen representation by only utilizing 1% and 10% of the labeled ImageNet's training data. As shown in Tab. 1b, our method outperforms previous approaches on 10% semi-supervised linear evaluation by a large margin. Higher accuracy is achieved if we further perform fine-tuning on this semi-supervised classification task. Details can be found in Appendix A.3.

Finally, we transfer the pretrained models to downstream tasks, *i.e.*, object detection and segmentation tasks on COCO (Lin et al., 2014). We also use Mask R-CNN (He et al., 2017) as in Sec. 5.1 by only replacing the backbone with ResNet18. As shown in Tab. 1c, our bounding box AP and mask AP both outperform the distillation-based methods. Our method also shows comparable results to the supervised baseline (*Supervised-IN.* entry in Tab. 1c), which relies on the labels of ImageNet dataset (Deng et al., 2009).

### 5.5   Discussion: Why the larger DNs show inferior Results?

In Sec. 5.1, we observe that the larger DNs show slightly inferior results to the individually pretrained counterparts under linear evaluation (Fig. 4), semi-supervised evaluation (Fig. 5) and downstream detection and segmentation tasks (Fig. 6). Here, we provide one possible explanation for the

| Methods | Teacher | | Student (ResNet18) | | |
|---|---|---|---|---|---|
| | Methods | Top1 | kNN | Top1 | Top5 |
| *Supervised* | | | | | |
| - | - | - | | 69.5 | |
| *Self-supervised* | | | | | |
| MoCo-v2 [8] | - | - | 36.7 | 52.2 | 77.6 |
| BYOL [15] | - | - | 50.2 | 60.8 | 83.6 |
| *Self-supervised with Distillation* | | | | | |
| SEED [12] | R50 (MoCo-v2[8]+200e) | 67.4 | 43.4 | 57.9 | 82.0 |
| | R50×2 (SwAV†[5]+400e) | 77.3 | 55.3 | 63.0 | 84.9 |
| CompRess [1] | R50 (MoCo-v2[8]+800e) | 70.8 | 53.5 | 62.6 | |
| | R50 (SwAV[5]+800e) | 75.6 | 56.0 | 65.6 | - |
| DisCo [13] | R50 (MoCo-v2[8]+200e) | 67.4 | - | 60.6 | 83.7 |
| | R50×2 (SwAV†[5]+400e) | 77.3 | - | 65.2 | **86.8** |
| Ours | expanded | - | **56.1** | **65.9** | 86.4 |

(a) **kNN and linear evaluation comparisons** on ImageNet. R50 is short for ResNet50 (He et al., 2016), and R50×2 represents a R50 at width 2×. † indicates using multi-crop strategy (Caron et al., 2020).

| Methods | 1% | | 10% | |
|---|---|---|---|---|
| (ResNet18) | Top1 | Top5 | Top1 | Top5 |
| MoCo-v2 [8] | 31.1 | 54.5 | 47.2 | 73.1 |
| BYOL [15] | 42.0 | 67.4 | 56.1 | 81.4 |
| SEED [12] | 43.5 | - | 54.9 | - |
| DisCo [13] | 47.5 | - | 54.6 | - |
| Ours | 47.1 | 71.8 | 59.7 | 82.0 |
| Ours (fine-tuning) | **49.2** | **73.7** | **62.5** | **83.6** |

| Methods | COCO Obj. Det. | | | COCO Inst. Segm. | | |
|---|---|---|---|---|---|---|
| (ResNet18) | $AP^{bb}$ | $AP^{bb}_{50}$ | $AP^{bb}_{75}$ | $AP^{mk}$ | $AP^{mk}_{50}$ | $AP^{mk}_{75}$ |
| *Random Init.* | 34.7 | 54.3 | 37.7 | 31.8 | 51.2 | 34.0 |
| *Supervised-IN.* | 37.1 | 57.3 | 40.0 | 34.0 | 54.3 | 36.4 |
| MoCo-v2 [8] | 36.6 | 56.7 | 39.7 | 33.4 | 53.8 | 35.8 |
| BYOL [15] | 36.8 | 57.1 | 39.7 | 33.6 | 54.2 | 35.7 |
| SEED [12] | 37.1 | 57.6 | 40.0 | 34.0 | 54.5 | **36.3** |
| CompRess [1] | 36.9 | 57.3 | 39.8 | 33.7 | 54.3 | 36.0 |
| Ours | **37.2** | **57.8** | **40.4** | **34.1** | **54.9** | **36.3** |

(b) **Semi-supervised evaluation** on ImageNet. Top1 and Top5 accuracy are reported.

(c) **COCO (Lin et al., 2014) object detection and segmentation** based on Mask R-CNN He et al. (2017).

Table 1: **Comparisons with SSL methods with distillation.** We compare the smallest DN slimmed from our DSPNet with the models trained by SSL methods with distillation.

above observations by investigating the working mechanism of our DSPNet. We first consider the individually pretrained baseline with MobileNet v3 at width $1.0\times$. Built upon that, the introducing of optimizing sampled sub-networks for the online encoder during the pretraining of our DSPNet can be viewed as an additional regularization on the full-size online encoder. This urges the good representation to be compressed to the sub-networks. In other words, such a training scheme makes a compromise between slimmablity and good representation for the full-size online encoder. The former accounts for a good imitation for the DNs to the whole one, and the latter for the qualified target network as a teacher, both of which contribute to powerful DNs. Hence, smaller DNs benefit from the knowledge distillation from the teacher and achieve better performance than individually pretrained ones, while the largest one has to accommodate those DNs as its sub-networks, leading to slightly inferior results in the context of missing explicit supervision in SSL. The experimental results in Fig. 7 *(Right)* also support our conjecture: as the number of DNs increases, the regularization becomes stronger, and thus the larger DNs shows more inferior results. Nevertheless, even though the only few large DNs are discarded considering their inferior performance, the others are also preferable for various downstream tasks, and our method still has an advantage in the training cost. Besides, we also show that our DSPNet can sever as good initialization for dynamic neural networks in downstream tasks, *e.g.*, Slimmable Networks (Yu et al., 2018), in Appendix B.

## 6 CONCLUSIONS

In this paper, we have proposed Discriminative-SSL-based Slimmable Pretrained Networks (DSP-Net), which can be trained once and then slimmed to multiple sub-networks of various sizes, and experimental results show that each of them learns good representation and can serve as good initialization for downstream tasks, *e.g.*, semi-supervised classification, detection, and segmentation tasks. Our method perfectly meets the practical use where multiple pretrained networks of different sizes are demanded under various resource budgets, and large training cost can be reduced.

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
