# OpenReview forum: "DSPNet: Towards Slimmable Pretrained Networks based on Discriminative Self-supervised Learning"
_ICLR.cc/2023/Conference — Submitted to ICLR 2023_

### Official Review · Reviewer_P1zK · 2022-10-22

**Confidence:** 4
**Clarity, Quality, Novelty And Reproducibility:** The quality of the paper does not mee…
**Correctness:** 2
**Technical Novelty And Significance:** 2
**Empirical Novelty And Significance:** 2
**Recommendation:** 3

**Strength And Weaknesses:**

Pros:

The method is rather straightforward.

Cons:

1. Limited novelty. This paper is just a straightforward application of slimmable networks with BYOL, namely, a combination of existing works (slimmable networks + BYOL).

2. Improper conclusion. In `Sec. Introduction`, the author claims `each of which learns good representation`. However, according to the experimental results, the performance of the largest network is worse than the individually trained network. See Figure 4 and Figure 5. The phenomenon is more obvious in Figure 8, near 2% gap between the performance of DSPNet and individually trained network (ResNet-34).
(2.1.) Besides, these individual results are provided by the authors. How about DSPNet compare with standard ResNet-50 trained individually?

3. Why only show results of ResNet-18 in Sec. 5.4? ResNet-34 with DisCo is 68.1%. ResNet-34 with BINGO[1] is 69.1. For ResNet-18, BINGO[1] also get 65.9% top-1 accuracy on ImageNet. The performance of DSPNet cannot surpass previous distillation methods while DSPNet also adopts distillation during training.

4. Combine point 2 and point 3, for large networks, DSPNet is worse than individually trained methods; for small networks, DSPNet is no better than previous distillation based methods. One advantage of DSPNet is that it can train for one go and get several networks. However, we have to accept its aforementioned disadvantages at the same time. Sometimes, a large drop in performance (~2% in Figure 8) is unacceptable, and it means we sacrifice a large model to obtain better small models. In this case, distillation based methods maybe more practical.

5. Figure 4 vs. Figure 8, why the largest network drops more when using ResNet. Could the authors provide some explanations? In Figure 8, why ResNet-34 has nearly 50M parameters? A ResNet-50 only has 25M parameters. Please check the Figure 8.

[1] BAG OF INSTANCES AGGREGATION BOOSTS SELF-SUPERVISED DISTILLATION. ICLR 2022.


**Summary Of The Paper:**

This paper proposes to combine the slimmable networks with BYOL to get multiple pre-trained networks in one go during training.

**Summary Of The Review:**

Due to the limited novelty and my suspicion of the practicability of the method, I vote for rejection.

---

### Official Review · Reviewer_xwkC · 2022-10-24

**Confidence:** 4
**Correctness:** 3
**Technical Novelty And Significance:** 2
**Empirical Novelty And Significance:** 3
**Recommendation:** 3

**Clarity, Quality, Novelty And Reproducibility:**

Clarity: Good

Quality: Fair

Novelty: Poor

Reproducibility: Good



**Strength And Weaknesses:**

Strength:
1. The motivation of this paper is clear and the method is simple to follow.
2. The experiments show some promising results.

Weakness:
1. I think this paper has limited technical novelty. From my perspective, this paper is just a direct combination of BYOL and Slimmable Network. On the basis of BYOL's loss, it is only necessary to introduce the sampling strategy in slimmable networks. The explanation given by the authors is that the difference is the use of a momentum teacher, but this is not the difference between self-supervised learning and supervised learning. There are also many self-supervised methods that do not require a momentum encoder.
2. In Figure 1, the authors present sub-networks with different widths as well as different depths. However, from the implementation details in Section 5.1, it seems that only the width is changed, and the depth is not involved. Hence, I think Figure 1 is misleading.
3. Why choose BYOL as your method, can other self-supervised methods work?
4. Missing related work [1].
[1] DATA: Domain-aware and task-aware self-supervised learning. CVPR2022.

**Summary Of The Paper:**

This paper proposes to train slimmable networks in the context of self-supervised learning, which can be trained once and then slimmed to multiple sub-networks of various sizes. The authors use BYOL as the baseline method and adopt sandwich rule and in-place distillation to sample & train sub-networks. Experimental results show that the performance of the largest model is lower compared to the separately trained model, and the performance of the small sub-networks can be improved. The authors also show that the pretrained models also generalize well on downstream detection and segmentation tasks.


**Summary Of The Review:**

I agree with the motivation of this paper but my major concern is the technical novelty of this paper. I don't see any challenges when combining BYOL and slimmable networks, nor does the author seem to have made any technical tweaks and optimizations.

If the authors can address my concerns, I am willing to raise my score.

---

### Official Review · Reviewer_ssnC · 2022-10-24

**Confidence:** 4
**Correctness:** 3
**Technical Novelty And Significance:** 3
**Empirical Novelty And Significance:** 3
**Recommendation:** 6

**Clarity, Quality, Novelty And Reproducibility:**

The paper is well-written and easy to follow. The idea of using slimmable network for pre-training different networks in a single pipeline and in a self-supervised manner is novel based on my knowledge.

**Strength And Weaknesses:**

Strengths:

- In my opinion, self-supervised learning considering the resource budget and efficient network architectures is interesting and applicable in real-world scenarios.
- The paper is well-written and easy to follow.
- The idea of training self-supervised representation learning with slimmable network that can be trained once and then slimmed to multiple subnetworks with different resource budgets is novel.
- The results show the effectiveness of the proposed approach in different downstream tasks.

Weaknesses:

1- In Table 1 (a), it would be interesting to see the results of one of the knowledge distillation-based approaches such as SEED with a teacher trained using BYOL strategy to be more comparable with the current approach.

2- MobileNet_v3 is also another architecture that is hard to train individually in a self-supervised manner as mentioned in SEED and DisCo. While the authors did some efforts in reporting results on this architecture, it would be more interesting to see the results in comparison to the existing KD-based approaches such as SEED. For instance, in Fig. 4, the individual trained Mobilenet_v3 has an accuracy near 66% while in SEED or Disco it is a lot lower and when it has been trained using KD by considering ResNet50 as the teacher, it reaches ~65. I would like more clarification on the evaluations on MobileNet_v3.


**Summary Of The Paper:**

This paper proposes DSPNet which can be trained once and then slimmed to multiple sub-networks of various sizes (e.g., different widths and depths) suitable for various resource budgets. The experimental results show that each of these sub-networks learns relatively good representation compared to the individually pre-trained ones and can serve as initialization for downstream tasks, e.g., semi-supervised classification, detection, and segmentation tasks.

**Summary Of The Review:**

The idea of using slimmable network for pre-training different networks in a single pipeline and in a self-supervised manner is novel based on my knowledge. I also would like to see the authors' answers to my concerns in the weaknesses section.

---

### Official Review · Reviewer_NqU9 · 2022-10-26

**Confidence:** 5
**Correctness:** 3
**Technical Novelty And Significance:** 2
**Empirical Novelty And Significance:** 2
**Recommendation:** 5

**Clarity, Quality, Novelty And Reproducibility:**

The paper is clear and easy to follow. There may be limited technical contribution but combining slimmable network with self-supervised learning seems to be valid.

**Strength And Weaknesses:**

Strength
1. The idea of combining self-supervised learning method with slimmable networks is technically sound.
2. The method achieves better performance than individually trained models.
3. The paper is easy to follow.

Weakness
1. There may be limited technical novelty in the proposed method. It is basically training slimmable networks using a self-supervised learning (e.g., BYOL) method. But I think demonstrating its effectiveness is still good.
2. The paper used BYOL, but the major backbone is MobileNet v3, which is not used in BYOL. I am not sure why the author didn't use the backbones used in BYOL (e.g., ResNet-50). I think a better way to demonstrate the effectiveness is following the same setting (both backbone and training hyperparameters) but just apply the slimmable training. The baseline results in the paper are much lower than those in the original BYOL, thus I am not very convinced.
3. Fig. 1-3 show that we could sample sub-networks by width and depth, but the main results are all based on width sampling only. Why don't use both width and depth in the main results?
4. I am not sure about the training cost shown in Fig.4. For example, for 0.5x width individually trained model, the time is 0.75x. But 0.5x width will reduce the computational cost of the model to around 0.25x.  I understand there could be some other time cost like data loading, but 0.75x still seems a lot to me for 0.5x width network.
5. In Tab 1(a), I don't think the comparisons to SEED, CompRess and DisCo are fair because they are using different self-supervised learning method.

**Summary Of The Paper:**

This paper proposed a self-supervised learning method to train a slimmable network. The self-supervised pre-trained model can be used to fit different computing resource for downstream tasks. The basic idea is to sample sub-networks following slimmable network and train these networks using a self-supervised learning method, e.g., BYOL. The proposed method is demonstrated to achieve comparable or better performance than individually self-supervised trained models, especially when the model is small.

**Summary Of The Review:**

I have some concerns about the technical contributino of this paper, but I think showing the effectiveness of combining slimmable network with self-supervised learning can still be a good contribution. My main concern is about the experimental results as explained in the weakness part.

---

### Decision · Program_Chairs · 2023-01-20

**Decision:**

Reject

**Justification For Why Not Higher Score:**

All the reviewers raised questions about the work. However, the authors have provided no response to any of the reviewers.

**Justification For Why Not Lower Score:**

N/A

**Metareview: Summary, Strengths And Weaknesses:**

*Summary*: The paper proposes a single network that can be trained using self-supervised learning and then slimmed down to networks of different sizes depending on the downstream requirements.

*Strengths*: (1) Interesting and technically sound idea. (2) The empirical results show that the slimmable networks perform better than similar capacity models trained individually.

*Weaknesses*: (1) The experimental results in this work rely on using a MobileNet architecture that wasn't originally used with the BYOL self-supervised method. While this in itself is not an issue, the baseline results reported for training this model are surprisingly low (as also noted by Reviewer ssnC & NqU9).  (3) It is not clear how the training cost is being computed and why it is limited to width reduction.